# The Effect of Physical Activity on Motor Skills of Children with Autism Spectrum Disorder: A Meta-Analysis

**DOI:** 10.3390/ijerph192114081

**Published:** 2022-10-28

**Authors:** Carlos Eduardo Monteiro, Elirez Da Silva, Ravini Sodré, Frederico Costa, André Soares Trindade, Priscila Bunn, Gabriel Costa e Silva, Fabrízio Di Masi, Estélio Dantas

**Affiliations:** 1Postgraduate Program in Nursing and Biosciences, University Federal of the State of Rio de Janeiro, Rio de Janeiro 22290-250, Brazil; 2Postgraduate Program in Exercise and Sport Science, University of the State of Rio de Janeiro, Rio de Janeiro 20550-013, Brazil; 3Department of Physical Education, Tiradentes University, Aracaju 49032-390, Brazil; 4Department of Physical Education, Faculty of Clube Nautico Mogiano, Mogi das Cruzes 08773-000, Brazil; 5Laboratory of Human Moviment Science, Colégio Pedro II, Rio de Janeiro 20080-001, Brazil; 6Laboratory of Physiology and Human Performance, University Federal Rural of the State of Rio de Janeiro, Rio de Janeiro 23897-215, Brazil

**Keywords:** neurology, motor coordination, physical exercise

## Abstract

Objective: The present study was aimed at analyzing the effect of physical activity on motor coordination in children with ASD. Methods: On 28 June 2021, a systematic review with meta-analysis was performed using the following databases: MEDLINE, SciELO, LILACS, PEDro, SPORTDiscus, CINAHL, SCOPUS, Web of Science, Cochrane, and Science Direct. We analyzed the methodological quality and risk of bias using the Jadad scale and Cochrane tool, respectively. Motor coordination results were meta-analyzed using the RevMan program. Two independent researchers used the Grading of Recommendations Assessment, Development, and Evaluation (GRADE) tool to assess the level of evidence from the meta-analysis. Results: We found four studies in the listed databases and five randomized clinical trials were included in the meta-analysis that included 109 children with ASD. Children with ASD who performed physical activity did not present significantly better motor coordination than control children (*p* = 0.12). Conclusions: Considering the clinical importance of physical activity for children with ASD, this systematic review with meta-analysis showed that physical activity had no statistically significant effects on coordination in individuals with ASD.

## 1. Introduction

Autism spectrum disorder (ASD) is not an illness; it can be characterized by deficits in reciprocal communication and social interaction, as well as restricted and repetitive patterns (stereotypes) of behavior, interests, or activities [1]. These symptoms are present from childhood and impair activities of daily living (ADLs) [2], as well as important motor experiences for development [2,3], directly reflected in the performance of motor tasks [4]. Many of these limitations are accentuated due to societal misunderstanding of these children. Being aware of one’s body domain in different spheres during childhood, in addition to interacting with the environment in which one is located, contribute to improving basic motor skills, including understanding difficult tasks, planning, and motor sequencing [3,5]. Reasons for problems with engaging in physical activity could be due to a poor offering of adaptive options, lack of qualification of teachers and coaches, and insufficient structure of clubs and schools. Parental support is extremely important in overcoming adversity and providing physically active behavior for children with ASD [6,7].

A recent study identified that boys aged 6 to 11 years with ASD engaged in physical activity less frequently or were less engaged (including in sports programs) compared with peers of the same age in the general population [8]. The condition makes it difficult to perform physical activities and maintain motor development. Cognitive dysfunction interferes in executive function regarding planning and motor action for physical activity. This may include delays in motor skills, which is not a determinant for diagnosis, but involves important deficits in body balance, global and fine motor coordination, and gait, among other abilities [9].

Epidemiological data show that the incidence of ASD in the world is about 1 in every 160 children [10], with the rate steadily increasing over the last five decades [11]. The main studies in disease control and prevention in the United States indicate that 1 in 54 people has autism [12]. In Brazil, there are no official statistics on the number of people with ASD. However, unofficial data from Paula et.al. [13] show that 1 in 360 people have this disorder, with the majority concentrated in the southeast part of the country [14], specifically in São Paulo. The number of diagnosed ASD cases has increased, especially after the Diagnostic and Statistical Manual of Mental Disorders (DSM) was updated in 2013 (fifth edition), with changes that promoted better understanding about the diagnosis [15]. This motivated more societal engagement and more interest from the scientific community regarding ASD [16]. Conducting more studies directed at this population is crucial to improve the quality of life of people with ASD. Motor coordination presents as a common dysfunction, compromising children’s performance of ADLs and school activities [17]. This subject has drawn the attention of researchers in the past.

Fournier et al. [18], comparing children and adults with ASD with control children and adults, concluded that ASD is associated with significant generalized changes in motor performance. Eight years later, Healy et al. [19] analyzed 29 experimental studies and concluded that engaging in physical activity contributes to improvements in the motor, manipulative, and fitness skills of young people with ASD. However, a limitation of this meta-analysis was the different types of experimental studies and different comparison groups. Shortly after, despite some methodological limitations in the studies selection, Huang et al. [20] performed an extremely important meta-analysis of four studies to verify the effect of physical activity on the sports ability of children and young people with autism, controlling the comparison groups (all were inactive controls) and study types (all were randomized controlled studies). 

Initially, when considering the four studies that included the outcome of interest, the results from Huang et al. [20] pointed to a lack of effect of physical activity on sports ability. After excluding two of the four studies [9,21] and an additional study [22] because they examined different age groups, the results pointed to the effectiveness of physical activity. However, the study by Pan et al. [23], one of the two remaining studies, included the age group from 6 to 12 years, whereas in the study by Sarabzadeh et al. [22], a similar age group was excluded. In addition, there was not enough strength of evidence to make a conclusion, as they only included two studies, with 19 participants in the experimental group and 19 in the control group. Two RCTs, by Hassani et al. [24] and Arabi et al. [25], were not included in the meta-analysis by Huang et al. [20] as they were published later. 

Thus, the aim of the present study was to analyze the effectiveness of physical activity on motor skills in children with ASD through a new meta-analysis.

## 2. Materials and Methods

The present work was based on criteria recommended by Preferred Reporting Items for Systematic Reviews and Meta-Analyses (PRISMA) [26] and registered in the Open Science Framework (https://osf.io/ufwxs accessed 25 June 2021).

### 2.1. Eligibility Criteria

The PICOS strategy was used to establish the eligibility criteria [27]. This meta-analysis included studies with participants who received a diagnosis of ASD between 3 and 18 years of age and performed any physical activity, compared with an inactive control group (participants who did not perform any type of physical activity). The outcome we evaluated was motor skills and the type of study included was randomized controlled trials.

### 2.2. Search Strategy

A systematic search was conducted on 28 June 2021, in the following databases: MEDLINE (US National Library of Medicine), Scientific Electronic Library Online (SciELO), Latin American and Caribbean Literature in Health Sciences (LILACS), Physiotherapy Evidence Database (PEDro), SPORTDiscus, CINAHL (Cumulative Index to Nursing and Allied Health Literature), SCOPUS, Web of Science, Cochrane, and Science Direct. The search phrases were developed using the Boolean logic operators “OR” between synonyms and “AND” among descriptors. The terms “motor skill” AND “autism spectrum disorder” were identified in the DeCS and MeSH descriptors and inserted as main descriptors for the search strategy, with the following filters: last 10 years, human, children from 0 to 18 years of age, and experimental applied studies.

### 2.3. Selection of Studies

The preliminary analysis was performed by two independent evaluators. Duplicates were removed, then studies were selected by title and abstract. Upon complete reading, studies were evaluated for eligibility. Any disagreements were resolved by consensus or decided by a third evaluator.

### 2.4. Evaluation of the Methodological Quality of Studies

A methodological evaluation of the randomized controlled studies was performed using the Jadad scale by two independent experienced researchers. Disagreements were resolved by a third evaluator. The following methodological criteria were investigated for each study: (1a) Was the study described as randomized? (1b) Was the randomization performed appropriately? (2a) Was there double-blinding? (2b) Was blinding performed correctly? (3) Was there a description of the sample losses [28]?

### 2.5. Assessment of the Risk of Bias in Studies

Cochrane’s tool (RoB 2.0) was used to assess the risk of conditional random field (CRF) bias in the selected studies by two independent experienced researchers. Disagreements were resolved by a third evaluator. The following items of risk of bias were evaluated: randomized sequence generation, secret allocation, blinding of patients, outcome evaluators and responsibility for data analysis, incomplete outcomes, selective reporting of outcomes, and other sources of bias [29]. RoB 2.0 is available at: https://sites.google.com/site/riskofbiastool/welcome/rob-2-0-tool (accessed 30 June 2021)

### 2.6. Data Collection Process

The following data were extracted: number of participants in experimental and control groups (*n*), age, physical activity performed, type of test used to measure motor skills, and test results.

### 2.7. Data Analysis

The motor skill results were obtained by a meta-analysis using the RevMan program (version 5.5; Cochrane Collaboration, 2015; http://cochrane-handbook.org, accessed 5 July 2021). Motor skill was considered an interval variable. The statistical method of inverse variance was used, and the standardized mean difference was used as a measure of effect, as motor skills were measured using different scales. The random effects analysis model was adopted to meet the significant I^2^, with the confidence interval equal to 95% for included studies, the meta-analysis, and the studies sorted out by contribution weight.

### 2.8. Evaluation of Level of Evidence of Meta-Analysis

Two independent researchers used the Grading of Recommendations Assessment, Development, and Evaluation (GRADE) tool [30,31] to evaluate the level of evidence of the meta-analysis. Disagreements were resolved by a third evaluator. There were four levels of classification to assess the quality of the evidence: high, moderate, low, and very low. Experimental studies start from high-quality evidence, whereas observational studies start from low-quality evidence. Five circumstances can lower the quality of evidence: risk of bias, inconsistency, indirect evidence, inaccuracy, and publication bias. Three elements can increase the quality of evidence: effect size, dose–response gradient, and control of confounding variables.

## 3. Results

Figure 1 shows a flow chart of the search for studies. An initial search returned 1771 articles. Then after excluding duplicate records, there were 1585 studies. After the eligibility criteria and exclusion criteria were met, based on different study types or outcome, different or combined interventions and populations, incomplete data, and placebo, there were four articles.

Table 1 briefly describes the selected studies in the present meta-analysis, including methodological design and statistical results.

The present meta-analysis assessed the methodological quality of the selected studies based on the following questions: (1) Was the study defined as random? (2) Was the randomization method adequate? (3) Was the study double-blind? (4) Was the masking method adequate? (5) Were there descriptions of losses and exclusions? According to these criteria, the assessment of the studies showed good methodological quality, with all studies adequately randomized. One of the four studies was not double-blinded. Another one of the four studies presented an adequate masking method, but did not describe losses or exclusions. 

Based on the randomization process, deviations from intended interventions, possible missing outcome data and their measurements, and selection of the reported results, the present meta-analysis demonstrated a low bias risk. 

Figure 2 and Figure 3 show a graphic display of estimated results in the selected studies, along with the overall results and suspected publication bias. In relation to the effectiveness of physical activity on motor skills in children with ASD, we observed a high heterogeneity (Tau^2^ = 7.48; Chi^2^ = 67.42, df = 4 (*p* < 0.00001); I^2^ = 94%). In addition, there was no publication bias in this meta-analysis (*p* = 0.7712).

Figure 4 classifies the level of evidence of selected studies according to the Grading of Recommendations Assessment, Development, and Evaluation (GRADE) approach.

## 4. Discussion

The objective of this systematic review with meta-analysis was to verify the effect of physical activity on motor skills in children with ASD. In the analyzed studies, 56 children with ASD who performed some type of physical activity did not present significantly different effects in motor skills compared with 53 control children (Figure 3). However, the number of studies (and participants) was much too small in the present meta-analysis, with a Cohen’s d value of 1.99, which is more than double the value considered as a large effect size (0.8) [32], representing great clinical relevance. Additionally, the analyzed studies reported significant increases in motor skills for individuals with ASD through physical activity.

On the other hand, the meta-analysis by Utesch et al. [33] analyzed more than 15,000 subjects, and showed significant gains in motor competence of children without ASD who performed physical activities. The authors demonstrated moderate-to-large positive associations between motor competence and physical fitness from early childhood to early adulthood. It is assumed that the reason for the difference in results between that meta-analysis and ours is the lower engagement in physical activity by children with ASD compared with non-ASD children [8]. The fragility of the executive function of children with ASD, who tend to be more impoverished in their ability to plan and perform motor activities, may have interfered in their engagement in physical activity, as reported in the studies [34]. The low level of physical fitness found in this population may be a further limiting factor in the search for studies on physical activity [35] and contributed to the results found in this study. Reinders et.al. [36] demonstrated the importance of physical activity for individuals with ASD in a very interesting literature review. The authors analyzed 40 studies and observed that physical activity had a mutual relationship with the social functioning of these individuals.

Teixeira [37] recommended sports practice as a treatment approach for people with ASD. This may directly favor engagement in the practice of physical activity, but difficulty playing with a group and lack of interest in offered objects, which are characteristics of people with ASD, may decrease their interest in playing sports [38]. Executive functions have a strong relationship with sensory processing in children with ASD regarding motor skills; these children show a weakness in attending to stimuli, whether visual, auditory, or tactile, in response to requested motor activities [39]. This can hinder their engagement in sports activities, as there is a relationship between cognitive and motor skills, corresponding to the planning and execution of actions related to play, recreation, and sports [23].

Possible deficits in executive function directly interfere with cognitive performance, especially in metacognition, which highlights the ability to manage and control tasks to be performed [40]. This contributes to sensory motor operation. Deficits in executive function cause poor motor performance, especially in children with more severe autism [41].

In addition to the lack of engagement in physical activities and poor offering of adaptive options in general due to ASD, another factor that may have contributed to the lack of effect of physical activity on motor skills in children with ASD in this study was the high heterogeneity of the meta-analysis (Figure 2). The heterogeneity among the included RCTs was likely due to the different types of physical activities performed: school physical education, Tai chi, team sports education, and table tennis (Table 1). In addition, variations in intervention time, weekly frequency, and duration of sessions may also have contributed (Table 1). Statistically, it was not possible to explain the cause of this high heterogeneity, because the number of RCTs on the subject was very small (Figure 1); thus, the small number of participants in this meta-analysis (Figure 2) may have contributed to an inaccurate result.

Selecting fewer than 10 studies can cause a loss of power in the funnel plot, as recommended by Cochrane [42]; however, it is a test that brings subjectivity to the results, and for this, the Egger test is more objective. The RCTs included in this meta-analysis presented a risk of low bias, and consequently, good methodological quality. In addition, there was no publication bias in this meta-analysis (Figure 3). Despite our efforts to achieve high scientific rigor in the planning and execution of this meta-analysis, the level of evidence of the meta-analysis was very low. Further studies on this subject could contribute to increasing the number of participants and developing similar physical activity protocols, which would reduce the inconsistency and inaccuracy (Figure 4).

In this meta-analysis, the motor coordination of children with ASD was evaluated by several tests: BOTMP, TGMD-2, and M-ABC-2. The BOTMP is intended to evaluate children who need special attention for motor development [43]. The TGMD-2 evaluates the performance of fundamental motor skills without distinguishing any pathology [44], and the M-ABC-2 assesses lags in motor coordination [45]. Ketcheson et al. [4] drew attention to the need for a more in-depth analysis of the fundamental motor skills of children with ASD. Considering the previous points, there is a need for an evaluation of motor coordination focused on people with ASD that considers all of their characteristics and allows for better understanding of their low engagement in physical activity.

Conducting further RCTs in the future will allow for the analysis of subgroups engaging in the same type of physical activity and result in an increase in the number of participants, which would contribute to more reliable and accurate results on the effect of physical activity on motor skills in children with ASD.

It is also suggested that RCTs be conducted to study methods to reduce the attention deficit of children with ASD, to improve their engagement in physical activities and to develop specific tests to measure the motor skills of these children.

## 5. Conclusions

Considering the clinical importance of physical activity for children with ASD and their motor skill development, this systematic review with meta-analysis showed that the practice of physical activity had no statistically significant effect on their motor skills. The small sample size in the analyzed studies did not allow for a definitive conclusion; a greater number of participants will be required in future studies. It is important to develop controlled physical exercise programs with specific motor tasks for better neuromotor development in ASD individuals.

However, this conclusion should be viewed with caution due to the lack of methodological standardization in the selected studies.

## Figures and Tables

**Figure 1 ijerph-19-14081-f001:**
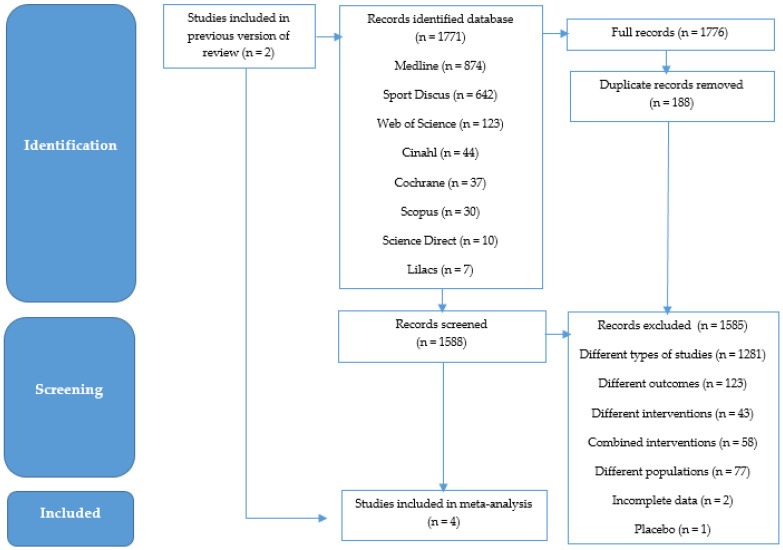
Flow chart of study search. Four randomized controlled trials were included, but meta-analyses were performed on five studies, because the study by Hassani [24] used two experimental groups performing physical activities.

**Figure 2 ijerph-19-14081-f002:**
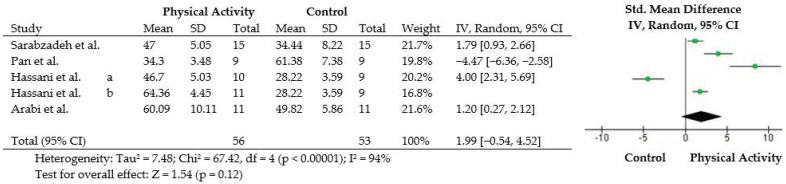
Forest plot of results of motor skills for children who did some kind of physical activity vs. controls. Four randomized controlled trials were included, but five results were included in meta-analysis, because the study by Hassani [24] used two experimental groups performing physical activities.

**Figure 3 ijerph-19-14081-f003:**
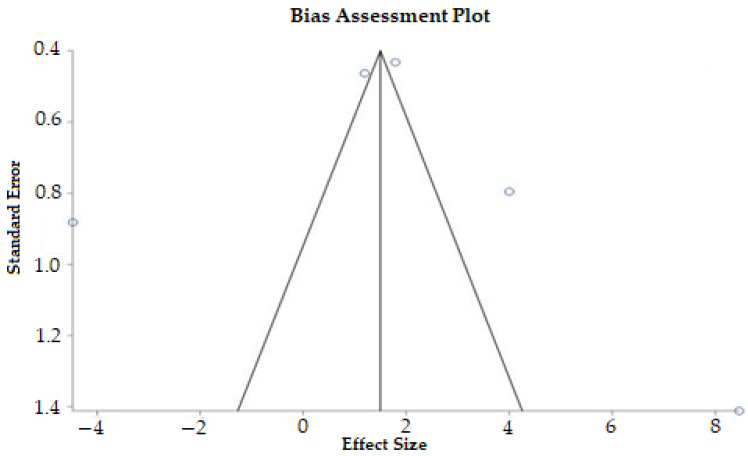
Funnel plot and Egger test to verify suspected publication bias. Egger: bias = 1.922 (95% CI = −17.303329 to 21.14733), *p* = 0.7712. Four randomized controlled trials were included, but five results were included in meta-analysis, because the study by Hassani [24] used two experimental groups performing physical activities.

**Figure 4 ijerph-19-14081-f004:**
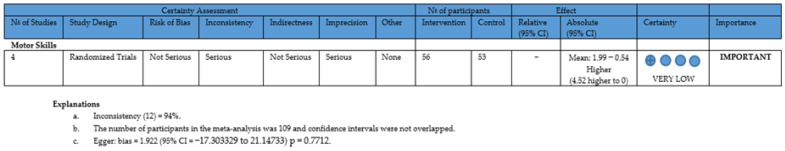
Level of evidence of meta-analysis. CI, confidence interval.

**Table 1 ijerph-19-14081-t001:** Common data extracted from included studies.

Author	Groups	Protocol	Motor Test	Results
Sarabzadeh et al. [22]	EG: Tai chi (*n* = 9) (8.88 ± 1.76 years) CG: Inactive (*n* = 9) (8.22 ± 1.92 years)	EG: 60 min Tai chi training program, 3 days a week for 6 weeks CG: No training	Movement Assessment Battery for Children (M-ABC-2)	EG = 34.30 ± 3.48 CG = 61.38 ± 7.38 *p* < 0.001
Pan et al. [23]	EG: Table tennis (*n* = 11) (9.68 ± 1.61 years) CG: Inactive (*n* = 11) (8.49 ± 1.76 years)	EG: 5 min warm-up, followed by table tennis technique for 20 min for motor skills, 20 min for motor skills associated with executive function, 20 min group game, 5 min warm-up; 70 min per session, 2 times per week for 12 weeksCG: No training	BOTMP	EG = 60.09 ± 10.11 CG = 49.82 ± 5.86 *p* < 0.01
Hassani et al. [24]	EGSPARK (*n* = 10) (9.10 ± 0.87 years) EGICPL (*n* = 11) (8.55 ± 0.68 years) CG: Inactive (*n* = 9) (8.70 ± 0.70 years)	EGSPARK: 16 sessions, 60 min twice weekly, followed by classes (10 min warm-up, 40 min motor skills, 10 min cool-down) EGICLP: 16 sessions, 60 min twice weekly, followed by classes at school (first phase of each session, talked about topics such as animals, favorite colors, and exercises) CG: No training	Bruininks–Oseretsky Test of Motor Proficiency (BOTMP)	EGSPARK = 46.70 ± 5.03 EGICPL = 64.36 ± 4.45 CG = 28.22 ± 3.59 Both programs significantly increased motor skills *p* < 0.005
Arabi et al. [25]	EG: Sports game education (*n* = 15) (8.44 ± 1.94 years) CG: Inactive (*n* = 15) (8.40 ± 2.01 years)	EG: 10 min warm-up with aerobic exercises (running, jumping), 30 min motor skill exercises (throwing, catching, kicking) with different sports games, 10 min warm-up; 50 min per session, 3 times per week for 10 weeks CG: No training	Test of Gross Motor Development (TGMD-2)	EG = 47 ± 5.05 CG = 34.44 ± 8.22 *p* = 0.001

CG, control group; EG, experimental group; RCT, randomized controlled trial; EGSPARK, experimental group of sport, play, and active recreation for children; EGICPL, experimental group of basic physical activity.

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
