# Peer review of "The Effect of Physical Activity on Motor Skills of Children with Autism Spectrum Disorder: A Meta-Analysis"

_ijerph, 2022, doi:10.3390/ijerph192114081_

Round 1
Reviewer 1 Report
The research problem is scientifically interesting and important for practice. The text was prepared with integrity. I highly evaluate the construction of the research project. The project is described accurately. The results are discussed factually. The content contains apt references to the literature. The structure of the text is correct. I do not formulate any comments.
Reviewer 2 Report
The results should be more developed and described because there are no clear.
Reviewer 3 Report
The paper presents a meta-analysis to investigate the effect of physical activity on motor skills in children with ASD. The results show that the practice of physical activity has no significant effect on coordination in individuals with ASD.
The paper is organized and written well.
I have one major concern as follow:
The study has been done on a limited number of subjects. Therefore, I think a large cohort of subjects needs to be studied to come up with such a conclusion.
Reviewer 4 Report
As I first read the title of the manuscript, I got very curious. Effects of Physical Activity on motor skills of children with ASD sounds very interesting for a researcher who is both keen on sports and special needs. But the study has so many weaknesses that I cannot recommend it for publication.
Concerning the formal aspects there has to be done a complete revision of most of the references in the text. You followed the guidelines of the journal up to page 2 (btw. where are the line numbers?), then you started by mixing the journal style (with numbers according to the references in parentheses) with the more common apa-style (reference presented as authors in parentheses). Additionally the reference list contains several entries, which have to be deleted (from number 44 on…)
But much more than the formal aspects, which can be addressed easily, the content and methodological aspects are not convincing.
If you are thinking about a new submission you have at least to put more effort in explaining ASD more in depth, especially the connection between the underlying causes of this kind of disability (ASD is NOT an illness!) and the motor coordination deficits. As a researcher in special needs, I would strongly encourage you not to argue in such a deficit-oriented way about ASD. Considering the state of art in disability studies (which is not the main focus in this study, I admit), the influence of society on people with ASD and their behaviour has to be highlighted much more. Some reasons for problems in engaging in physical activity are surely the poor offer of adapted possibilities, the missing qualification of coaches or the insufficient structure of clubs. These aspects have to be kept in mind, when arguing about the reasons of poor engagement, as you did in your discussion. Is not all because of the disability itself…
Concerning the methodological approach, I don’t really understand your aim. Did you want to add evidence to further meta-analyses like Healy et al. (2018) and Huang et al. (2020)? Then I really don’t get, why you only included 4 studies. What about the studies included by Huang et al. (2020)? You did not explain clearly, why you added the study of Sarabzeh et al (2019), which was excluded by the former meta-analysis.
Another big concern I have with the results section of your manuscript. It only consists of tables and figures, without a single written paragraph. Tables have to speak for themselves, but you have to explain the results and put them into context. Additionally the tables should have been done with more effort, concerning design and unity. In Table 1 for example, you present the same coordination test for the studies of Hassani et al. (2020) and Pan et al. (2020) but present different abbreviations (BOTMP vs. BOT 2). Are these different tests? Another example: The statistical significant difference in the Hassani study presented (p<.0.005) belongs to what groups? GESPARK vs. GEICPL? GESPARK vs. GC? GEICPL vs GC? (btw. please use English abbreviations like CG for control group to make it easier for non Portuguese speaking colleagues).
The discussion has several severe limitations as well. As you stated correctly, the number of studies (and participants) is much too small for a meta-analysis. Why did you do it anyway? A systematic review, which should have taken into account the strengths and weaknesses of the four studies, would have been the better way, in my opinion. The use of different motor tests, as mentioned correctly by the authors, is another hint for NOT doing a meta-analysis in that way you did it.
On page 8 you have literally echoed many aspects that have been described before. Why? It seems that a final review of the text did not take place.
In summary, this is a manuscript that deals with a very exciting and relevant topic area. Unfortunately, both the methodological implementation and the theoretical embedding and relevance of the results for the target group are not competent enough.
Round 2
Reviewer 2 Report
The article has improved fllowing the reviewer´s comments.
Reviewer 4 Report
see file attached
